# IL-23 signaling is not an important driver of liver inflammation and fibrosis in murine non-alcoholic steatohepatitis models

Jose E. Heredia[1], Clara Sorenson[2], Sean Flanagan[2], Victor Nunez[2], Charles Jones[2], Angela Martzall[2], Laurie Leong[2], Andres Paler Martinez[1], Alexis Scherl[2], Hans D. Brightbill[3], Nico Ghilardi[1¤], Ning Ding[1]*

1 Department of Discovery Immunology, Genentech, South San Francisco, CA, United States of America, 2 Department of Pathology, Genentech, South San Francisco, CA, United States of America, 3 Department of Translational Immunology, Genentech, South San Francisco, CA, United States of America

¤ Current address: Dice Therapeutics, South San Francisco, CA, United States of America
* ding.ning@gene.com

**Data Availability Statement:** All relevant data are within the paper and its Supporting Information files.

## Abstract

Non-alcoholic fatty liver disease (NAFLD), represents an unmet medical need that can progress to non-alcoholic steatohepatitis (NASH), which, without intervention, can result in the development of cirrhosis and hepatocellular carcinoma (HCC). Inflammation is a pathological hallmark of NASH, and targeting key inflammatory mediators of NASH may lead to potential therapeutics for the disease. Herein, we aimed to investigate the role of IL-23 signaling in NASH progression in murine models. We showed that recombinant IL-23 can promote IL-17 producing cell expansion in the liver and that these cells are predominately γδ T cells and Mucosal Associated Invariant T cells (MAITs). Reciprocally, we found that IL-23 signaling is necessary for the expansion of γδ T cells and MAIT cells in the western diet (WD) diet induced NASH model. However, we did not observe any significant differences in liver inflammation and fibrosis between wild type and *Il23r-/-* mice in the same NASH model. Furthermore, we found that *Il23r* deletion does not impact liver inflammation and fibrosis in the choline-deficient, L-amino acid-defined and high-fat diet (CDA-HFD) induced NASH model. Based on these findings, we therefore propose that IL-23 signaling is not necessary for NASH pathogenesis in preclinical models and targeting this pathway alone may not be an effective therapeutic approach to ameliorate the disease progression in NASH patients.

## Introduction

Non-alcoholic fatty liver disease (NAFLD) is defined as a chronic liver disease that imparts excess lipid accumulation in the liver without secondary causes such as viral infections or alcohol abuse [1,2]. NAFLD can progress from simple hepatic steatosis to non-alcoholic steatohepatitis (NASH) characterized by inflammation and fibrosis, which is a significant risk factor for cirrhosis and hepatocellular carcinoma (HCC) [3–6]. Within the past decades, the epidemic of obesity has led to the sharp rise of NALFD/NASH incidence [2,7]. However, there are no FDA-approved therapies for NASH driven chronic liver disease, which may be largely

**Funding:** The author(s) received no specific funding for this work.

**Competing interests:** All authors are or were employees of Genentech, a member of the Roche group, and may hold Roche stock or stock options.

due to our limited understanding of molecular underpinnings of liver inflammation and fibrosis.

IL-23 is a cytokine that has been implicated in IL-17 driven pathologies such psoriasis, colitis, and autoimmune diseases [8–11]. As an IL-12 cytokine family member, IL-23 is composed of a heterodimer of IL-12p40 subunit and IL-23p19 subunit (IL-23p19/p40) which signals through the IL23R and IL12Rβ1 dimeric receptor [12]. Mice that lack IL-23p19 demonstrate an inability to drive the expansion of pathogenic IL-17 producing cells [13–15]. All IL-17 expressing cells also express RAR-related orphan receptor gamma t (RORγt), the master transcription factor driving the differentiation of IL-17 producing T cells [16,17]. In this regard, previous studies have implicated that IL-17 producing cells promote liver inflammation and fibrosis [18]. It has also been reported that there is accumulation of IL-17 producing cells in the livers from NASH patients or diet induced NASH murine models [19–22]. Additionally, IL-17 has been shown to be elevated by hepatotoxic agents such as carbon tetrachloride and Concanavalin A in models of acute hepatitis [18,20–23]. While these studies suggest that IL-17 producing T cells may contribute to a pro-inflammatory milieu, which can predispose to chronic liver disease, the role of IL-23 signaling in NASH pathogenesis has not been fully dissected.

In this study, we hypothesized that IL-23 signaling may play an important role in NASH pathogenesis. We showed that systemic administration of recombinant IL-23 protein induces IL-17 producing cell expansion in the liver and that these cells are predominately γδ T cells and Mucosal Associated Invariant T cells (MAITs). Reciprocally, genetic ablation of *Il23r* attenuates γδ T and MAIT cell expansion in western diet (WD)-induced NASH model. However, we did not observe that *Il23r* deletion reduces liver inflammation and fibrosis or improves liver function in the same model. Similarly, we found that *Il23r-/-* mice are not protected from liver inflammation and fibrosis in another model, the choline-deficient, L-amino acid-defined and high-fat diet (CDA-HFD) induced NASH model. Thus, these results do not support a causal role of IL-23 signaling in NASH pathogenesis and suggest that targeting IL-23 signaling alone may not be a viable therapeutic strategy to treat NASH patients.

## Materials and methods

### Mouse studies

All animal experiments were performed after approval from the Institutional Animal Care and Use Committee (IACUC) of Genentech. *Il23r-/-* mice were generated as described previously [24], the control group was littermate wild type (WT) mice. Diet used in this study was purchased from Research Diets; normal diet (ND) was compared either to Western Diet (WD) (cat#D19021501) composed of 40% kcal fat, 22% kcal fructose, and 1.25% cholesterol or choline deficient L-amino acid derived high fat diet (CDA-HFD) (cat#A06071302) composed of 60% kcal fat, 0.1% methionine, and no added choline. All mice were fed with diet starting at 8 weeks of age, and all mice used were males. The ND and WD cohorts were challenged with diet for 20 weeks. The ND and CDA-HFD cohorts were challenged with diets for 9 weeks. C57BL/6J mice from Jackson Laboratory were used for intraperitoneal injection (IP) with PBS or recombinant IL-23. Recombinant murine IL-23 was purchase from R&D. Mice were injected with either PBS or 0.5ug recombinant IL-23 for three consecutive days and livers were harvested 24hrs after last injection.

### Liver digestion and flow cytometry

Upon $CO_2$ euthanasia, serum was collected, and livers were perfused with 1X PBS, via portal vein. Livers were collected for either histology, or RNA extraction, or tissue processing for

non-parenchymal cell (NPC) isolation. All liver samples were processed at the same time by transferring the livers in c-tubes (Miltenyi) and adding 5mL of digestion media consisting of 0.2% Collagenase Type 2 (Worthington), 0.1% DNAse I (Roche), 1% BSA (Sigma), in RPMI media. Samples were digested using MACS Miltenyi dissociator followed by incubation at 37 degree for 30mins in shaker at 120rpm. Samples were then centrifuged at 1600rpm for 5mins and resuspended in 1X PBS and passed through a 70um cell strainer. Samples were pelleted, and resuspend in 15% Percoll, centrifuged for 1600rpm for 15mins without brake. The pellets were NPC fraction, free of hepatocytes. NPCs were then resuspended in 1X PBS, stained with LIVE/DEAD fixable dye (Invitrogen) at a 1:1000 dilution, incubated on ice for 15mins, washed, resuspended in FACS buffer (PBS + 2.5mM EDTA + 5% BSA) with FcR block (Miltenyi), and stained with the appropriate conjugated fluro-antibodies. For RORgt staining, cells were processed and stained using the FOXP3 Transcription Factor staining kit (BD). For intracellular staining of IL-17A, NPCs were stimulated with leukocyte activation cocktail with GolgiPlug (BD) for 4hours in RPMI media. Then cells were washed, FcR blocked, and stained with appropriate antibodies. Samples were run and analyzed on Symphony analyzer (BD).

## Antibodies

Antibodies are listed: anti-CD45 (BD, 30-F11), anti-γδ chain receptor (BD, GL3), anti-TCRβ chain (BD, H57-597), anti-CD3e (BD, 145-2C11), anti-CD4 (BD, RM4-5), anti-CD8 (Biolegend, 53–6.7), anti-CD90 (BD, 53–2.1), anti-SiglecF (BD, E50-2440), anti-Ly6G (BD, 1A8), anti-Ly6c (BD, AL-21), anti-CD11c (BD, N418), anti-CD11b (BD, M1/70), anti-CD64 (Biolegend, X54-5/7.1), anti-IA/IE (BD, M5/114.15.2), anti-F4/80 (Biolegend, BM8).

## RNA extraction, reverse transcription and quantitative realtime PCR

RNA was isolated from approximately 100mg of liver tissue using 1mL Trizol using the bead homogenizer Qiagen method, followed by addition of 200uL chloroform, resuspended samples were centrifuged for 10mins at 13krpm, 300uL clear top aqueous layer was transferred to new tube followed by the addition of 300uL 70% Ethanol. The 600uL samples were then loaded on a RNeasy Mini purification column (Qiagen) for RNA isolation. RNA quantification and purity was analyzed with NanoDrop 2000 (Thermo Scientific). 1ug of RNA was used for cDNA synthesis using Iscript First Strand cDNA kit (BioRad). cDNA templates were combined with Taqman probes (Thermo), and Taqman Universal PCR Master Mix (Thermo), and run on QuantaStudio 6 Flex (Applied Biosystems). The cat# for Taqman probes used in qPCR are listed in Table 1.

**Table 1. Taqman probe information.**

| Gene | Taqman probe cat# |
|---|---|
| Rpl19 | Mm01606039_g1 |
| Cxcl2 | Mm00436450_m1 |
| Ccl2 | Mm00441242_m1 |
| Cd68 | Mm03047343_m1 |
| Cxcl10 | Mm00445235_m1 |
| Col1a1 | Mm00801666_g1 |
| Col3a1 | Mm01254476_m1 |
| Col1a2 | Mm00496696_g1 |

## Histology

Paraffin embedded liver tissues were sectioned for hematoxylin and eosin (H&E) or for trichrome staining. Automated image analysis was conducted on trichrome stained slides to assess fibrosis and inflammation. Features counted towards inflammation include inflammatory cells (lobular inflammation), primarily macrophages with some neutrophils, and areas of hepatocyte injury/ductular reaction.

## Serum biomarker and cytokine analysis

The liver chemistry panel consists of the following assays: Alanine Transaminase (ALT), Aspartate Transaminase (AST), Alkaline Phosphatase (AP), Albumin (ALB), and Triglycerides (TRIG). All assays were performed on the Beckman Coulter Au480 chemistry analyzer using the analytical principle of spectrophotometry and potentiometry. (Beckman Coulter Inc., Brea CA). Serum cytokines were measured using Luminex bead assay (Millipore platform).

## Quantification and statistical analysis

GraphPad Prism 6 was used for statistical analysis using the unpaired student t-test or one-way ANOVA. Statistical details are provided in the figure legends.

## Results

### Recombinant IL-23 increases RORγt cell accumulation in the liver

In order to determine whether IL-23 is sufficient to induce hepatic expansion of RORγt positive IL-17 producing cells, we intraperitoneally injected 0.5ug recombinant mouse IL-23 (rmIL-23) daily for three consecutive days into mice fed on normal diet (ND) and analyzed the livers 24hrs after the last injection. Administration of rmIL-23 led to a five-fold expansion of hepatic Ki67[+] RORγt cells (Fig 1A and 1B) and a two-fold increase in the percentage of Ki67[+] RORγt cells when compared to vehicle control (Fig 1C). RORγt cells that proliferated actively were identified as MAIT and γδ T Cells (Fig 1D and 1E). rmIL-23 treatment also induced an increase in the percentage of hepatic neutrophils and inflammatory monocytes when compared to vehicle control (Fig 1F and 1G). These results thus suggest that IL-23 is sufficient to induce RORγt cell accumulation and pro-inflammatory response in the liver.

### Western Diet induced hepatic expansion of RORγt cells is dependent on IL-23R

Next, we investigated whether IL-23 signaling is required for RORγt cell accumulation in the animal model of NASH. Western Diet (WD) consisting of high fat, high fructose, and added cholesterol have been established to induce several NASH phenotypes including hepatic inflammation, fibrosis and an increase in hepatocellular injury measured by the serum biomarkers such as alanine aminotransferase (ALT), aspartate aminotransferase (AST), all in a nutritional setting without liver damaging chemicals [25,26]. We fed WT and *Il23r-/-* mice with normal diet (ND) or WD for 20 weeks (Fig 2A) [27]. The WD induced the expansion of RORγt[+] γδ T cells and MAIT cells in the livers from WT mice (Fig 2B and 2C). In *Il23r-/-* mice, we found that WD-induced expansion of γδ T cells and MAIT cells were normalized to the baseline (Fig 2B and 2C). Furthermore, we observed a significant decrease of IL-17A production in γδ T cells from *Il23r-/-* NPCs compared to WT (Fig 2D). These data suggest that IL-23 plays an important role in regulating IL-17 producing cells in WD induced NASH model.

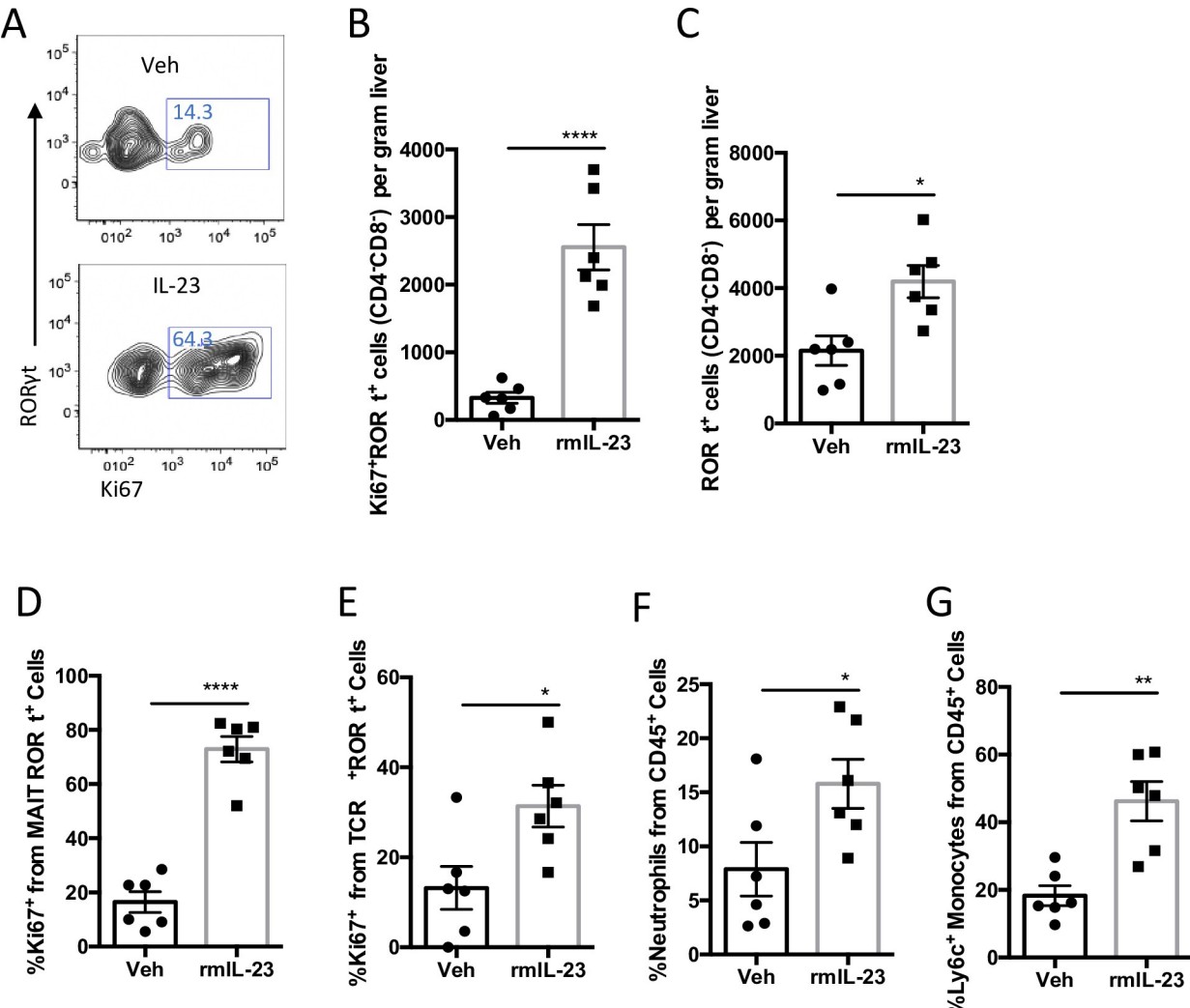

**Fig 1. rmIL-23 administration induces RORγt proliferation in liver.** FACS staining was performed in liver NPCs isolated from mice administered either PBS or recombinant mouse IL-23 (rmIL-23) by IP injections (A-G): Representative FACS gate, IL-23 induced greater frequency of RORγt⁺Ki67⁺ T Cells gated from CD3e⁺CD4⁻CD8⁻ (A). Total RORγt⁺Ki67⁺ T Cells, indication of proliferative cell, and total RORγt (CD4-CD3-) T Cells quantified in the liver (B-C). Percentage quantification of Ki67+ of RORγt⁺MAITs and RORγt⁺ γδ T Cells (D-E). Percentage of Neutrophils and Ly6c⁺ monocytes from CD45+ Cells (F-G). Groups: Vehicle (PBS): n = 6, rmIL-23 (3x 0.5ug): n = 6. Data represents mean ± S. D. *p < 0.05, **p < 0.001, ****p < 0.0001, two-tailed t-test.

## IL-23 signaling is not critical for liver inflammation and fibrosis induced by WD in mice

Having established its critical role in WD-indued RORγt cell accumulation, we explored the contribution of IL-23 signaling to WD-induced liver inflammation and fibrosis. While there is a clear increase of liver inflammation induced by WD, we did not observe any noticeable differences in liver inflammation between WT and *Il23r-/-* livers from WD fed mice as assessed by histology and pro-inflammatory gene expression (Fig 3A and 3B). On the other hand, while we observed a modest, but statistically significant, reduction of pro-inflammatory monocytes in the WD indued *Il23r-/-* liver (Fig 3C), there was no significant difference of neutrophil infiltration to the liver between Il23-/- and WT mice (Fig 3D), the main myeloid cell known to be recruited by IL-17 induced chemokines. In addition, we found that the serum levels of

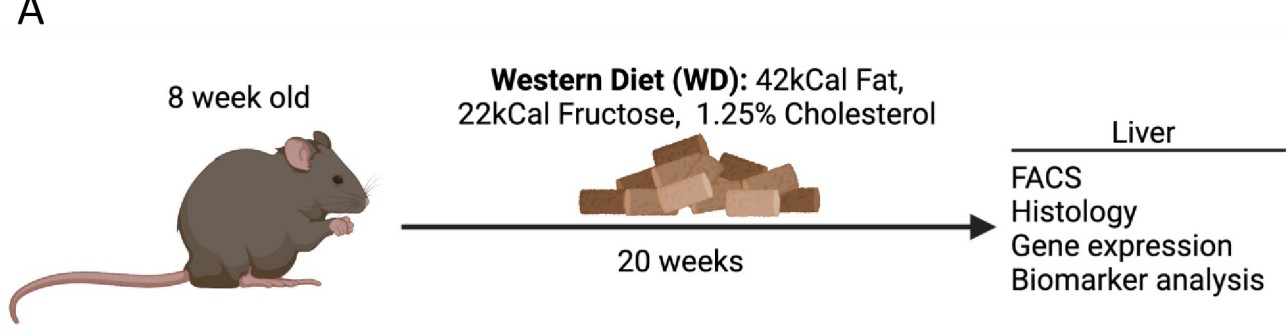

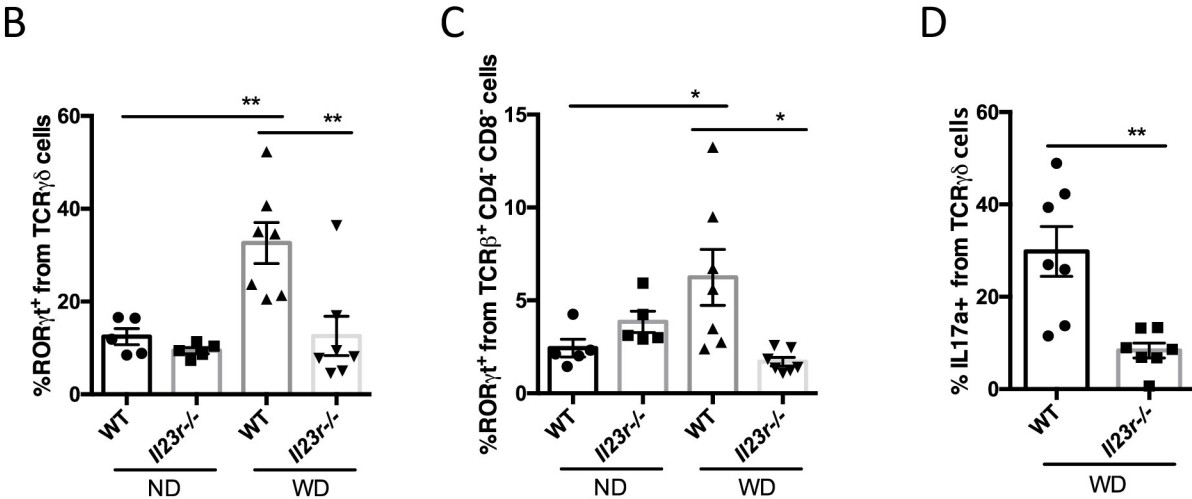

**Fig 2. IL-23 is required for WD-induced hepatic IL-17 producing cell expansion.** WT or *Il23r-/-* mice were fed a ND or WD for 20 weeks, followed by terminal analyses (A). FACS quantification of percent RORγt$^+$ from γδ T Cells and RORγt$^+$ from MAIT cells (CD3e$^+$TCRb$^+$CD4$^-$CD8$^-$) (B-C). Percentage of IL-17A positive cells in γδ T cells from WD fed WT and *Il23r-/-* liver NPCs stimulated with leukocyte activation cocktail with GolgiPlug (BD) for 4 hours (D). Groups: ND WT n = 5, ND *Il23r-/-* n = 5, WD WT n = 7, WD *Il23r-/-* n = 7. Data represents mean ± S.D. *p < 0.05, **p < 0.005, one-way ANOVA.

keratinocytes-derived chemokine (KC) (Fig 3E) and interferon gamma-induced protein 10 (IP-10) (Fig 3F) were not changed in *Il23r-/-* mice. Next, to address the role of IL-23 signaling in liver fibrosis in WD-induced NASH model, we evaluated hepatic collagen content by trichrome staining analysis and hepatic collagen gene expression. We did not observe a significant difference of collagen content at the histology level as well as at the transcriptional level (Fig 4A and 4B). Overall, these results do not support IL-23 signaling as the main driver of liver inflammation and fibrosis in WD-induced NASH model.

## IL-23 signaling does not contribute to WD induced liver dysfunction

Next, we examined the impact of *Il23r* deletion on liver function. In this regard, we measured several serum biomarkers of liver function. The results showed that there is little difference of serum ALT/AST, alkaline phosphatase (AP), albumin, cholesterol, and triglyceride levels

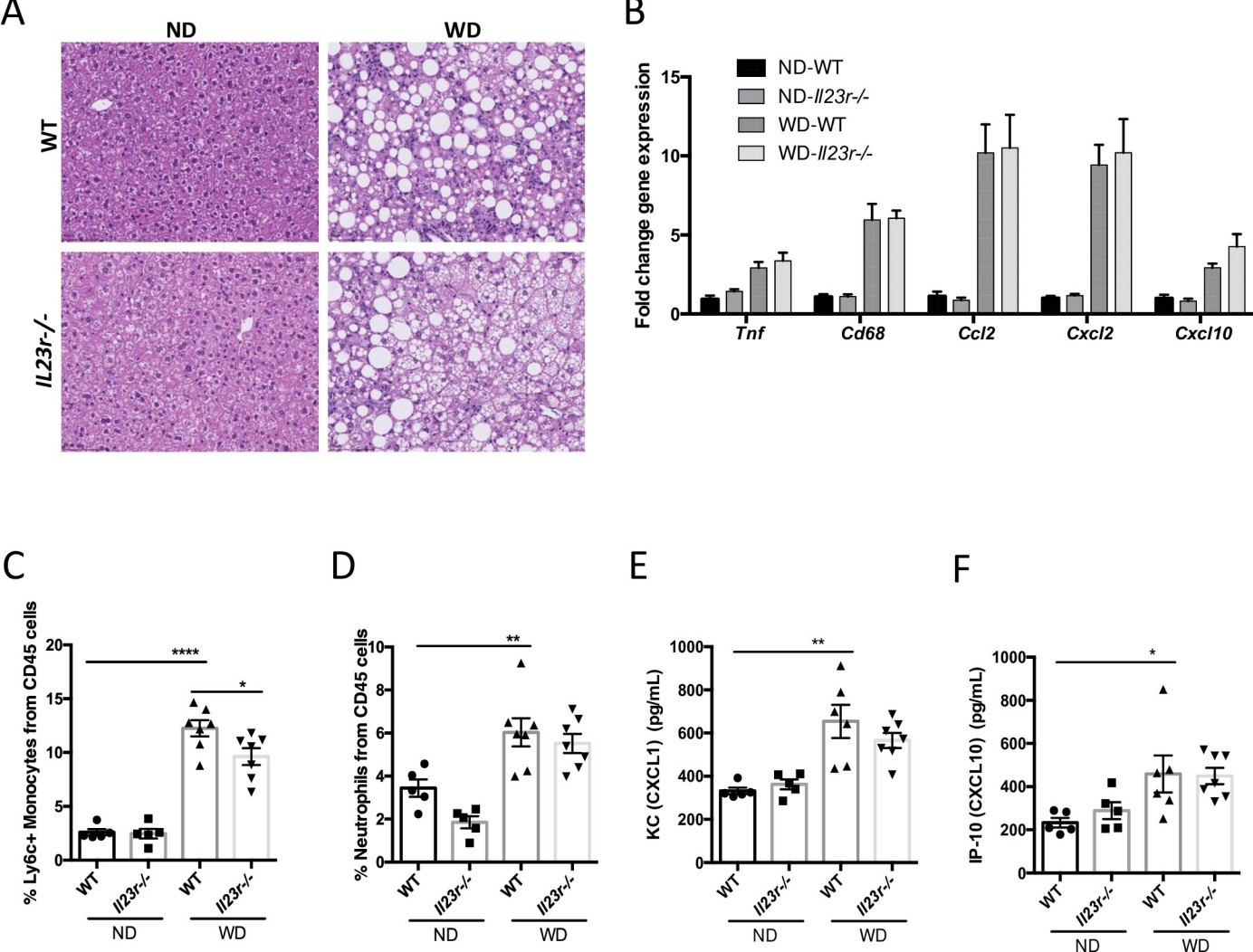

**Fig 3. IL-23 signaling is dispensable for WD-induced hepatic inflammation.** H&E staining images of livers from WT and *Il23r-/-* mice on ND or WD (A). Liver mRNA levels of *Tnf*, *Cd68*, *Ccl2*, *Cxcl2*, and *Cxcl10* (B). Liver FACS analysis of percent Ly6c$^+$ monocytes and neutrophils between groups. Luminex results for serum levels of KC (E) and IP-10 (F). Groups: ND WT n = 5, ND *Il23r-/-* n = 5, WD WT n = 7, WD *Il23r-/-* n = 7. Data represents mean ± S.D. $^*$p < 0.05, $^{**}$p < 0.005, $^{****}$p < 0.00005, one-way ANOVA.

between WT and *Il23r-/-* mice fed on WD (Fig 5C–5H). Similarly, IL23R deficiency appears not to affect the WD-induced whole-body weight as well as liver weight gains (Fig 5A and 5B). Collectively, these data suggests that IL-23 signaling may not contribute to liver dysfunction caused by WD-induced metabolic imbalance.

## IL-23 signaling does not contribute to liver inflammation and fibrosis in the CDA-HFD model of NASH

To complement our findings in WD-induced NASH model, we sought to determine whether IL-23 signaling contributes to NASH pathogenesis in another animal model. In this regard, we chose the CDA-HFD model (Fig 6A) because this model has been demonstrated to recapitulate steatosis, inflammation, and progressive fibrosis in the liver [28]. The CDA-HFD significantly induced the expansion of RORγ+ γδ T Cells and MAIT cells in the liver (Fig 6B and

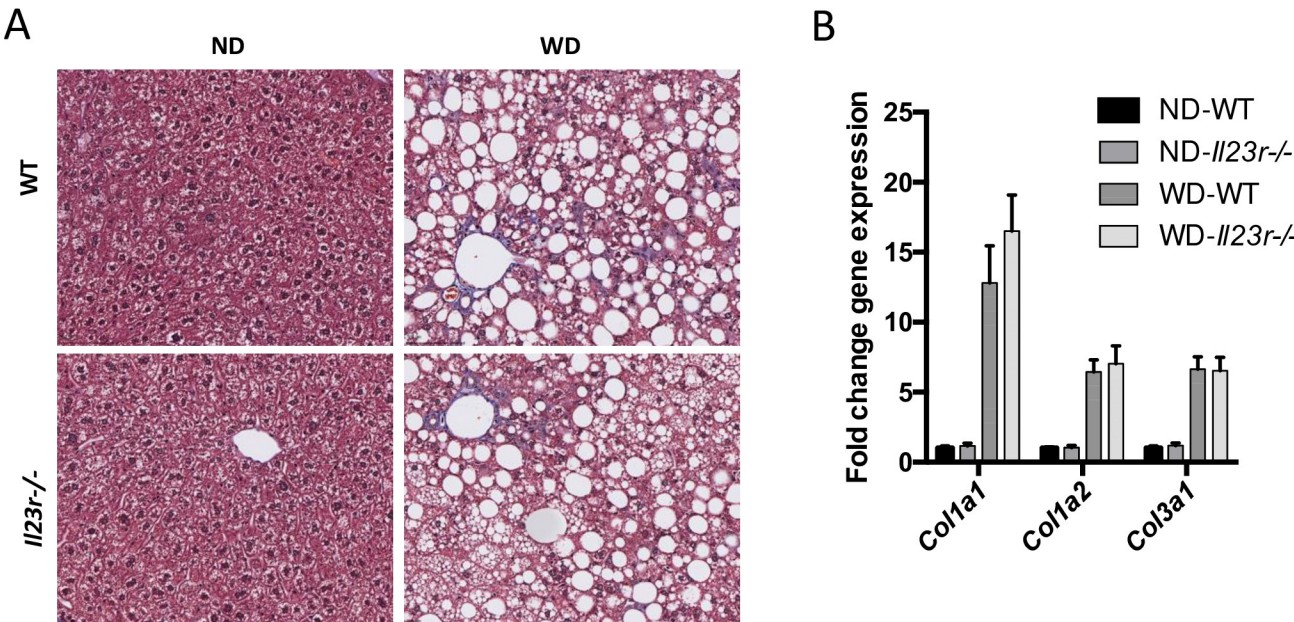

**Fig 4. IL-23 signaling does not contribute to WD-induced liver fibrosis.** Trichrome staining images of livers from WT and *Il23r*-/- mice on ND or WD (A). Liver mRNA levels of *Col1a1*, *Col1a2*, and *Col3a1*. Groups: ND WT n = 5, ND *Il23r*-/- n = 5, WD WT n = 7, WD *Il23r*-/- n = 7.

6C). Consistent to our observation in WD-induced NASH model, we found no impact of IL-23R depletion on liver inflammation and fibrosis by histology (Fig 6D). Similarly, we observed that *Il23r* deletion does not affect pro-inflammatory monocyte and neutrophil infiltration to liver induced by CDA-HFD treatment (Fig 6E and 6F). Furthermore, no significant changes in hepatic inflammatory or fibrogenic gene expression were detected between WT and *Il23r*-/- mice fed on CDA-HFD (Fig 6G). In addition, we did not observe any significant change in body weight, liver weight, and a variety of serum biomarkers for liver function between WT and *Il23r*-/- mice (Fig 7A–7H). Taken together, these results suggest that IL-23 signaling does not contribute to liver inflammation and fibrosis in CDA-HFD NASH model.

## Discussion

NAFLD/NASH is an unmet medical need that is increasingly common around the world. The incidence of NAFLD world-wide is approximately 25%, and the prevalence of NASH patients in NAFLD patients appears to be associated with biopsy status and regions. For example, the pooled global prevalence of NASH from biopsied NAFLD patients was estimated to be 59.1% while this ratio decreases to 7–30% in NAFLD patients without an indication for biopsy [7,29]. There are currently no approved therapies for NAFLD/NASH, and pro-inflammatory pathways have been proposed to be a class of appealing targets for this complex disease [30]. In this regard, it came to our attention that hepatic IL-17 producing cells have been shown to promote liver inflammation and dysfunction [18–21]. However, genetic dissection of this pathway, particularly its upstream regulator IL-23, in preclinical NASH models is lacking and the target candidacy of this IL-17/IL-23 axis in NASH is yet to be fully established.

In this context, we therefore chose to investigate the contribution of IL-23 signaling to NASH pathogenesis by testing IL23R deficient mice in animal models of NASH. Our data showed that, while recombinant IL-23 is sufficient to drive IL-17A producing cell expansion and pro-inflammatory myeloid cell infiltration in liver, *Il23r*-/- mice are not protected from

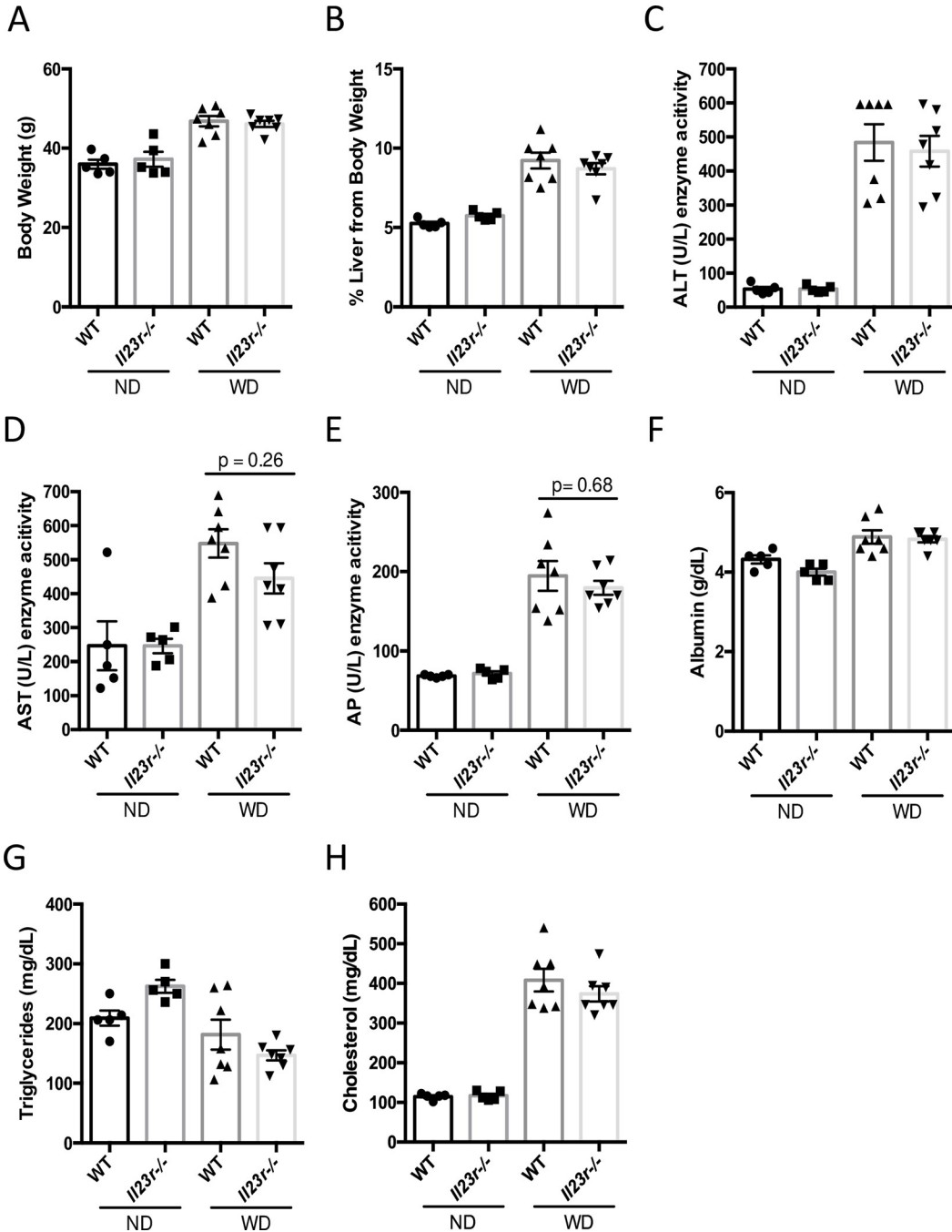

**Fig 5. IL-23 signaling does not contribute to WD-induced liver dysfunction.** Body weight (A) and percent liver weight to body weight (B). Measurement of serum liver enzymes for Alanine Aminotransferase (ALT) (C), Aspartate Aminotransferase (AST) (D), and Alkaline Phosphatase (AP) (E). Serum albumin protein (F), serum cholesterol (G) and serum triglycerides (H) were also measured. Groups: ND WT n = 5, ND *Il23r-/-* n = 5, WD WT n = 7, WD *Il23r-/-* n = 7. Data represents mean ± S.D, one-way ANOVA.

liver inflammation and fibrosis in two NASH models, suggesting the contribution of IL-23 signaling to NASH pathogenesis is minimal. These observations thus challenge the assumption that IL-17 producing cells that have been shown to be present in NASH patient liver samples

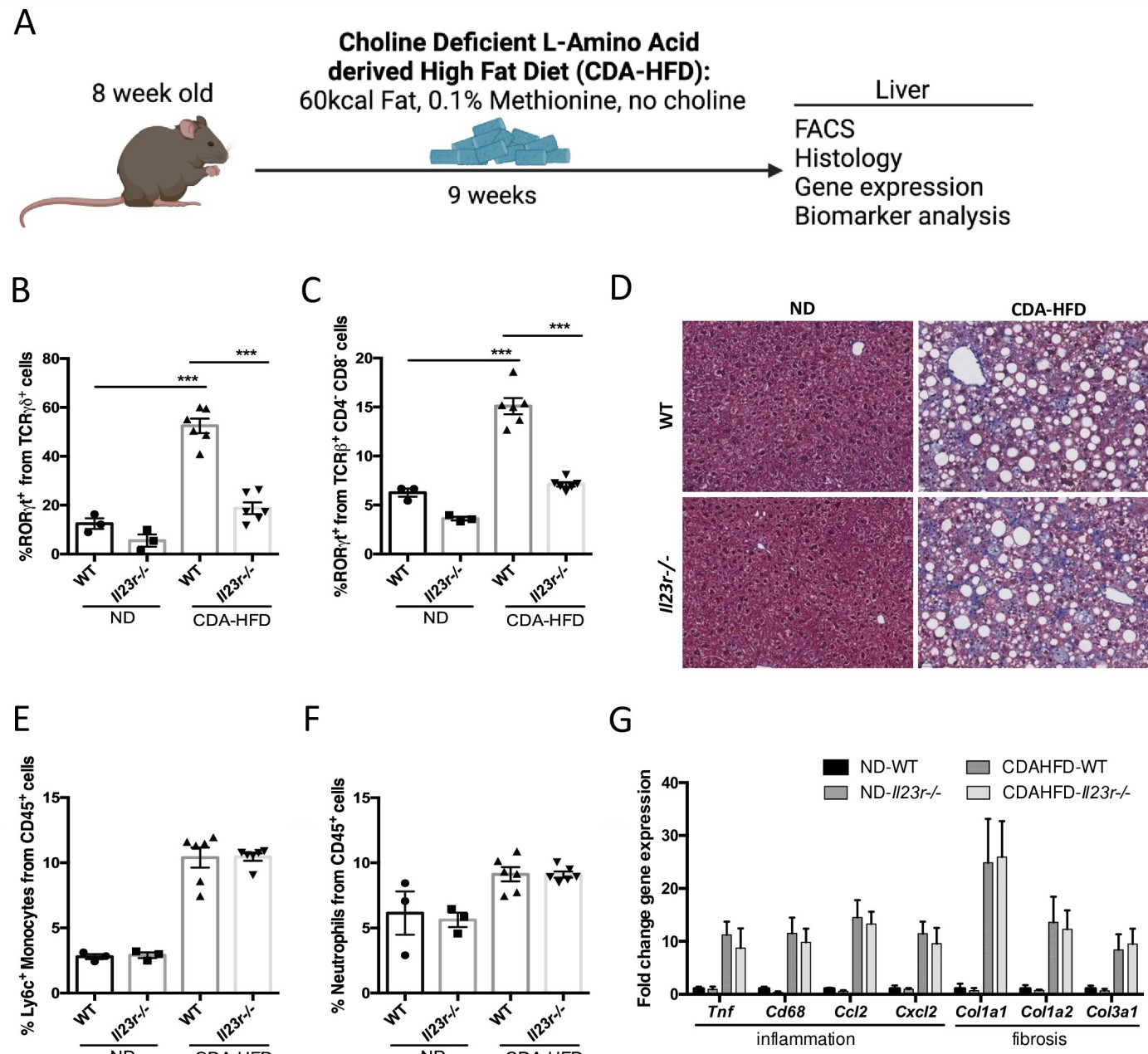

**Fig 6. IL-23 signaling does not contribute to liver inflammation and fibrosis in the CDA-HFD NASH model.** WT and *Il23r-/-* mice were fed CDA-HFD for 9 weeks, followed by liver analysis (A). Percent quantification of hepatic RORγt in γδ T Cells (B) and MAITs (C). Trichrome staining images and quantification of Inflammation (E) and Trichrome Collagen content (F). Liver mRNA expression of *Tnf*, *Cd68*, *Ccl2*, *Cxcl2*, *Col1a1*, *Col1a2*, and *Col3a1* (G). Groups: ND WT n = 3, ND *Il23r-/-* n = 3, CDA-HFD WT n = 6, CDA-HFD *Il23r-/-* n = 6. Data represents mean ± S.D. ***p < 0.0005, one-way ANOVA.

may play a causal role in the disease pathogenesis [31]. It should also be noted that we cannot rule out IL-23's contribution to non-NASH liver fibrosis as some reports suggest IL-23 signaling plays a role in cholestatic or viral driven liver fibrosis [18,32]. Nevertheless, the dispensability of IL-23 signaling in NASH driven liver inflammation is intriguing given its critical role in a wide variety of pro-inflammatory diseases. Since it is well documented that many inflammatory factors such as cytokines and PAMPs are elevated in NASH models, it is not inconceivable

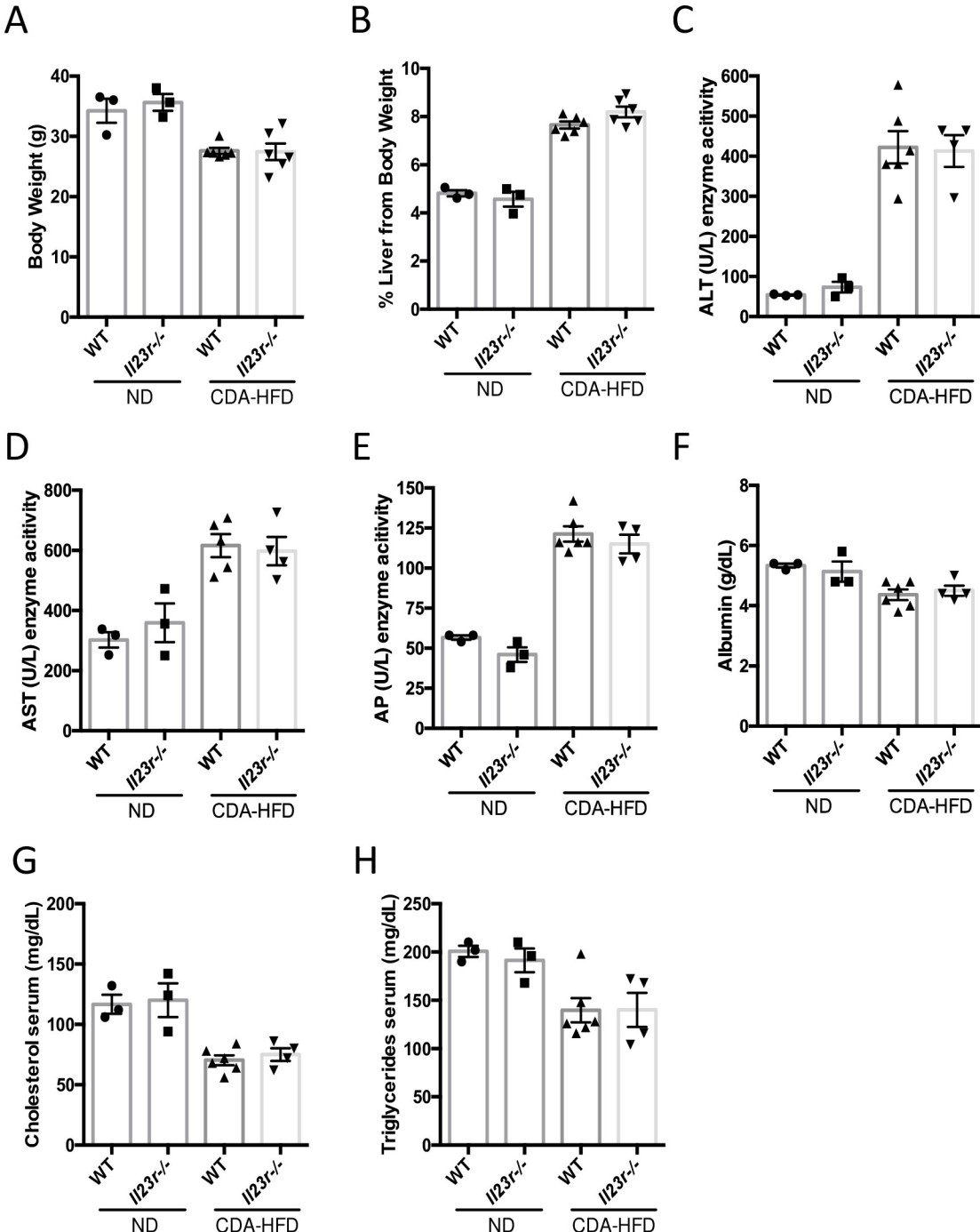

**Fig 7. IL-23 signaling does not contribute to CDA-HFD-induced liver dysfunction.** Body weights in gram (g) (A) and percent liver weights from body weight (B). Quantification of serum liver enzymes for Alanine Aminotransferase (ALT) (C), Aspartate Aminotransferase (AST) (D), and Alkaline Phosphatase (AP) (E) measured. Serum Albumin protein (F), serum cholesterol (G) and serum triglycerides (H) quantified. Groups: ND WT n = 3, ND *Il23r-/-* n = 3, CDA-HFD WT n = 6, CDA-HFD *Il23r-/-* n = 6. Data represents mean ± S.D.

that the accumulation of these factors may mask any effects of IL-23 in the NASH models [30,33]. Further studies are warranted to dissect the potential crosstalk between IL-23 and other proinflammatory cytokines during the pathogenesis of NASH.

While we examined the role of lL-23 signaling in two preclinical models of NASH, it should be noted that, currently, there is no single animal model that can perfectly replicate all disease features of human NASH patients [34]. For examples, WD diet model mimics metabolic profiles of human patients such as obesity, insulin resistance and inflammation but this model is unlikely to progress to advanced liver fibrosis (F3/4) unless animals are fed on the diet for an extended period of time (>25 weeks) [34,35]. On the other hand, CDA-HFD model elegantly develops progressive liver fibrosis and inflammation although it lacks the certain metabolic features of human NASH patients [28,34]. Therefore, by recognizing the limitations of preclinical models used in this manuscript, we cannot fully rule out the contribution of IL-23 signaling to NASH pathogenesis in a more human disease relevant setting. Future studies using improved animal models or human samples may be warrantied to test this hypothesis.

## Conclusions

In summary, we present the evidence that *Il23r-/-* mice are not protected from liver inflammation and fibrosis in two NASH preclinical models, thus suggesting that targeting IL-23 signaling alone may not be an effective therapeutic approach for NASH. Our study also supports the necessity of leveraging genetic models to validate drug targets when possible and suggests that the overall role of IL-23/IL-17 axis in NASH may need to be re-evaluated.

## Supporting information

**S1 File. Raw values of all graphs.**
(XLSX)

## Acknowledgments

We thank C.K. Poon and Terence Ho for FACS support; Ganesh Kolumam and Mark Chen for helpful discussion on NASH models.

## Author Contributions

**Conceptualization:** Jose E. Heredia, Ning Ding.

**Data curation:** Jose E. Heredia, Clara Sorenson, Sean Flanagan, Victor Nunez, Charles Jones, Angela Martzall, Laurie Leong, Andres Paler Martinez, Alexis Scherl, Hans D. Brightbill.

**Formal analysis:** Jose E. Heredia, Hans D. Brightbill.

**Investigation:** Jose E. Heredia, Clara Sorenson, Sean Flanagan, Victor Nunez, Charles Jones, Angela Martzall, Laurie Leong, Andres Paler Martinez, Alexis Scherl, Hans D. Brightbill.

**Methodology:** Clara Sorenson, Sean Flanagan, Victor Nunez, Charles Jones, Angela Martzall, Laurie Leong, Andres Paler Martinez, Alexis Scherl, Hans D. Brightbill.

**Project administration:** Nico Ghilardi.

**Resources:** Clara Sorenson, Sean Flanagan, Victor Nunez, Charles Jones, Angela Martzall, Laurie Leong, Andres Paler Martinez, Alexis Scherl, Hans D. Brightbill, Nico Ghilardi.

**Supervision:** Nico Ghilardi, Ning Ding.

**Visualization:** Nico Ghilardi.

**Writing – original draft:** Jose E. Heredia, Ning Ding.

**Writing – review & editing:** Jose E. Heredia, Ning Ding.

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
