## [Decision Letter · Decision Letter 0]

29 Jun 2022

PONE-D-22-09562IL23 signaling is not an important driver of liver inflammation and fibrosis in murine non-alcoholic steatohepatitis modelsPLOS ONE

Dear Dr. Ding,

Thank you for submitting your manuscript to PLOS ONE. After careful consideration, we feel that it has merit but does not fully meet PLOS ONE’s publication criteria as it currently stands. Therefore, we invite you to submit a revised version of the manuscript that addresses the points raised during the review process.

ACADEMIC EDITOR: Please address all comments from the reviewer.==============================

We look forward to receiving your revised manuscript.

Kind regards,

Aloysious Dominic Aravinthan, MBBS, FRCP, PhD

Academic Editor

PLOS ONE

Journal Requirements:

Additional Editor Comments (if provided):

A well written work and certainly worthy of publication once the reviewer's comments are addressed.

Reviewers' comments:

Reviewer's Responses to Questions

**Comments to the Author**

1. Is the manuscript technically sound, and do the data support the conclusions?

Reviewer #1: Yes

2. Has the statistical analysis been performed appropriately and rigorously? 

Reviewer #1: Yes

3. Have the authors made all data underlying the findings in their manuscript fully available?

Reviewer #1: Yes

4. Is the manuscript presented in an intelligible fashion and written in standard English?

Reviewer #1: Yes

5. Review Comments to the Author

Reviewer #1: Overall, this is a robust piece of work which adds to the scientific body of literature around NAFLD pathogenesis.

Comments as follows:

1/ Broad comment: I think it is important to mention the inherent problems in translating findings from pre-clinical models into the clinical realm i.e. lack of fidelity/concordance in terms of metabolic and histological progression. Therefore, while the overall signal from the data presented herein is that IL23 knockout mice are not protected from inflammation and fibrogenesis, this does not necessarily equate into IL23 signalling being unimportant in NAFLD pathogenesis. It may be more generous to state that targeting IL23 signalling ALONE is not sufficient as an effective therapeutic approach to NAFLD.

2/ Please pay attention throughout the manuscript to language -there are multiple instances of excess use of words like 'the' and 'of'. Examples include:

(A) Discussion line 147 ' the preclinical NASH models' (no need for the word 'the' here).

(B) Results line 83 'signalling is required for of RORyt' - no need for 'of' here.

(C) Introduction line 58 'an important role in the NASH pathogenesis' - no need for 'the' here.

3/ Discussion line 142. The authors quote the Zounossi meta-analysis to describe prevalence of NASH among NAFLD cohorts. I think using the figure of 59.1% is potentially misleading and I think it should be revised. The authors of the original meta-analysis themselves state that the prevalence of NASH among biopsied NAFLD cohorts is likely to be subject to selection and ascertainment bias (given that the biopsied patients were those with a high index of suspicion for steatohepatitis based on non-invasive parameters). It is probably more accurate to quote the 7-30% estimate given in the same article.

4/ Overall, I would suggest using more scientific language where possible, e.g.

(A) Introduction line 40 - the word 'occurence' is used. I suggest incidence.

(B) Introduction line 54 - liver 'damaging' agents' is used. I would suggest using the word toxins.

(C) Introduction line 63 - the word 'dramatic' is used. What do the authors mean by this? Can they quantify this? Can they use more scientific language?

(D) Results line 87- the phrase 'liver damage' is used - I suggest hepatocellular injury

(E) Results line 133 - the word 'fibrotic' is used to refer to gene expression. I suggest using the word 'fibrogenic'

6. PLOS authors have the option to publish the peer review history of their article (what does this mean?). If published, this will include your full peer review and any attached files.

Reviewer #1: No

---

## [Author Response · Author response to Decision Letter 0]

13 Jul 2022

Response to Reviewer #1 

We appreciate the constructive comments of Reviewer #1 and have now revised the manuscript to address these concerns.

1. Broad comment: I think it is important to mention the inherent problems in translating findings from pre-clinical models into the clinical realm i.e. lack of fidelity/concordance in terms of metabolic and histological progression. Therefore, while the overall signal from the data presented herein is that IL23 knockout mice are not protected from inflammation and fibrogenesis, this does not necessarily equate into IL23 signalling being unimportant in NAFLD pathogenesis. It may be more generous to state that targeting IL23 signalling ALONE is not sufficient as an effective therapeutic approach to NAFLD.

We are grateful for the reviewer's positive feedback on our manuscript, and we also agree with the reviewer that the gap between preclinical NASH models and human disease may increase the difficulty of translating animal data to human settings in a confident manner. To address this limitation of our manuscript, we added a specific discussion section on this issue in our revised manuscript and toned down our conclusion on the role of IL-23 signaling in NASH as suggested.

2. Please pay attention throughout the manuscript to language -there are multiple instances of excess use of words like 'the' and 'of'. Examples include:

(A) Discussion line 147 ' the preclinical NASH models' (no need for the word 'the' here).

(B) Results line 83 'signalling is required for of RORyt' - no need for 'of' here.

(C) Introduction line 58 'an important role in the NASH pathogenesis' - no need for 'the' here.

We apologize for these language issues. In the revision, we reduced the excess use of certain words including the ones mentioned by the reviewer.

3. Discussion line 142. The authors quote the Zounossi meta-analysis to describe prevalence of NASH among NAFLD cohorts. I think using the figure of 59.1% is potentially misleading and I think it should be revised. The authors of the original meta-analysis themselves state that the prevalence of NASH among biopsied NAFLD cohorts is likely to be subject to selection and ascertainment bias (given that the biopsied patients were those with a high index of suspicion for steatohepatitis based on non-invasive parameters). It is probably more accurate to quote the 7-30% estimate given in the same article.

Thanks for the reviewer raising this important point and we agree with reviewer that this reference needs to be interpreted with an extra caution. We toned down this section and revised the ratio based on the suggestion.

4. Overall, I would suggest using more scientific language where possible, e.g.

(A) Introduction line 40 - the word 'occurence' is used. I suggest incidence.

(B) Introduction line 54 - liver 'damaging' agents' is used. I would suggest using the word toxins.

(C) Introduction line 63 - the word 'dramatic' is used. What do the authors mean by this? Can they quantify this? Can they use more scientific language?

(D) Results line 87- the phrase 'liver damage' is used - I suggest hepatocellular injury

(E) Results line 133 - the word 'fibrotic' is used to refer to gene expression. I suggest using the word 'fibrogenic'

We highly appreciate the reviewer's advice, and we revised the language as suggested.

---

## [Editor Report · Decision Letter 1]

31 Aug 2022

IL-23 signaling is not an important driver of liver inflammation and fibrosis in murine non-alcoholic steatohepatitis models

PONE-D-22-09562R1

Dear Dr. Ning Ding,

We’re pleased to inform you that your manuscript has been judged scientifically suitable for publication and will be formally accepted for publication once it meets all outstanding technical requirements.

Kind regards,

Aloysious D Aravinthan, MBBS, FRCP, PhD

Academic Editor

PLOS ONE

Additional Editor Comments (optional):

All comments raised by the reviewer has been addressed satisfactorily and the manuscript reads well.

---

## [Editor Report · Acceptance letter]

6 Sep 2022

PONE-D-22-09562R1 

IL-23 signaling is not an important driver of liver inflammation and fibrosis in murine non-alcoholic steatohepatitis models 

Dear Dr. Ding:

I'm pleased to inform you that your manuscript has been deemed suitable for publication in PLOS ONE. Congratulations! Your manuscript is now with our production department. 

Kind regards, 

on behalf of

Dr. Aloysious Dominic Aravinthan 

Academic Editor

PLOS ONE